# Elamipretide: A Review of Its Structure, Mechanism of Action, and Therapeutic Potential

**DOI:** 10.3390/ijms26030944

**Published:** 2025-01-23

**Authors:** Cheryl Tung, Fahimeh Varzideh, Emanuele Farroni, Pasquale Mone, Urna Kansakar, Stanislovas S. Jankauskas, Gaetano Santulli

**Affiliations:** 1Department of Medicine, (Division of Cardiology), Wilf Family Cardiovascular Research Institute, Fleischer Institute for Diabetes and Metabolism (FIDAM), Einstein Institute for Aging Research, Albert Einstein College of Medicine, New York City, NY 10461, USA; cheryl.tang@einsteinmed.edu (C.T.); Emanuele.farroni@einsteinmed.edu (E.F.); pasquale.mone@einsteinmed.edu (P.M.); urna.kansakar@einsteinmed.edu (U.K.); stanislovasjankauskas@einsteinmed.edu (S.S.J.); 2Department of Molecular Pharmacology, Einstein-Mount Sinai Diabetes Research Center (ES-DRC), Einstein Institute for Neuroimmunology and Inflammation (INI), Albert Einstein College of Medicine, New York City, NY 10461, USA; fahimeh.varzideh@einsteinmed.edu; 3Montevergine Clinic, Mercogliano, 83013 Avellino, Italy

**Keywords:** bioenergetics, cardiolipins, clinical trials, Elamipretide, heart failure, mitochondria

## Abstract

Mitochondria serve an essential metabolic and energetic role in cellular activity, and their dysfunction has been implicated in a wide range of disorders, including cardiovascular conditions, neurodegenerative disorders, and metabolic syndromes. Mitochondria-targeted therapies, such as Elamipretide (SS-31, MTP-131, Bendavia), have consequently emerged as a topic of scientific and clinical interest. Elamipretide has a unique structure allowing for uptake in a variety of cell types and highly selective mitochondrial targeting. This mitochondria-targeting tetrapeptide selectively binds cardiolipin (CL), a lipid found in the inner mitochondrial membrane, thus stabilizing mitochondrial cristae structure, reducing oxidative stress, and enhancing adenosine triphosphate (ATP) production. Preclinical studies have demonstrated the protective and restorative efficacy of Elamipretide in models of heart failure, neurodegeneration, ischemia–reperfusion injury, metabolic syndromes, and muscle atrophy and weakness. Clinical trials such as PROGRESS-HF, TAZPOWER, MMPOWER-3, and ReCLAIM elaborate on preclinical findings and highlight the significant therapeutic potential of Elamipretide. Further research may expand its application to other diseases involving mitochondrial dysfunction as well as investigate long-term efficacy and safety of the drug. The following review synthesizes current knowledge of the structure, mechanisms of action, and the promising therapeutic role of Elamipretide in stabilizing mitochondrial fitness, improving mitochondrial bioenergetics, and minimizing oxidative stress.

## 1. Introduction

Mitochondria are primarily responsible for cellular energy metabolism, serving as the main source of adenosine triphosphate (ATP) to perform essential biological functions. Additional roles of mitochondria include apoptosis regulation and the management of reactive oxygen species (ROS) [1,2]. Dysfunction in these processes regulated by mitochondria can lead to cellular damage, inflammation, and disease progression, making mitochondria an attractive target for therapeutic intervention [3]. In addition to impaired energy production, mitochondrial dysfunction can result in reduced cellular calcium buffering, the generation of free radicals, and the activation of proteases, phospholipases, and nitric oxide synthase, potentially culminating in apoptotic or necrotic cell death [4]. Mitochondrial dysfunction has been identified in numerous chronic diseases, including diabetes, ischemic heart disease, heart failure, acute and chronic kidney diseases, and neurodegenerative diseases such as Parkinson’s disease and Alzheimer’s disease [2,3,4]. Furthermore, the toxicity of drugs such as doxorubicin may be attributed to mitochondrial dysfunction [3]. Therapies targeting mitochondria and addressing mitochondrial dysfunction are consequently of interest.

Elamipretide (SS-31, MTP-131, Bendavia) is a mitochondria-targeting tetrapeptide designed to treat mitochondrial dysfunction via binding to cardiolipin (CL) on the inner mitochondrial membrane [3,5]. Cardiolipin is a critical phospholipid that maintains mitochondrial structure and electron transport chain function [3]. Consequently, the interaction of Elamipretide with cardiolipin allows for the stabilization of mitochondrial cristae, a reduction in oxidative stress, and the enhancement of ATP production, cumulatively addressing key aspects of mitochondrial dysfunction [1]. The therapeutic potential of Elamipretide has been explored across a range of diseases associated with mitochondrial dysfunction [6,7,8,9,10,11,12,13,14]. Preclinical studies have demonstrated its efficacy in minimizing cardiac and renal ischemia–reperfusion injury, protecting against neurodegenerative diseases, attenuating muscle atrophy and weakness, and preventing the development of metabolic syndromes [2]. Clinical trials have further highlighted its potential, particularly in heart failure, Barth syndrome, and primary mitochondrial myopathy. However, variability in clinical outcomes underscores the need for continued research to optimize its application.

This review provides a comprehensive overview of the structure, mechanism of action, and therapeutic potential of Elamipretide. By synthesizing findings from preclinical and clinical studies, we aim to evaluate its current role and future prospects in treating mitochondrial dysfunction.

## 2. Structure of Elamipretide

Elamipretide is a synthetic tetrapeptide with the sequence (D-Arg-Dmt-Lys-Phe-NH2) [2]. Its structure (Figure 1) is designed to target the mitochondria and interact with cardiolipin, a key phospholipid found in the inner mitochondrial membrane.

The unique design of Elamipretide allows it to penetrate cells, accumulate in the mitochondria, and stabilize mitochondrial structure and function [3].

Elamipretide has two positively charged amino acids (D-Arg and Lys), which allow it to preferentially target the inner mitochondrial membrane, which has a high negatively charged cardiolipin content. The electrostatic attraction between the positive charges of the peptide and negatively charged cardiolipin facilitates the accumulation of Elamipretide within the mitochondria [1,3]. Interestingly, despite the presence of the positively charged arginine and lysine residues, Elamipretide retains a high cell permeability, which is suggested to be due to positive charge shielding by electrons in π orbitals in aromatic rings of phenylalanine and dimethyltyrosine (Dmt) [15]. Dmt also contributes to the stabilization of the peptide structure by preventing oxidation, adding to its overall chemical stability [2]. Furthermore, the amphipathic nature of Elamipretide (Figure 2), due to both hydrophobic (Phe and Dmt) and hydrophilic (D-Arg and Lys) regions, allows it to interact with both the lipid bilayer and the aqueous environment within cells, promoting its mitochondrial targeting and membrane penetration [16].

The combination of hydrophobic interactions (through Dmt and Phe) and electrostatic interactions (through D-Arg and Lys) allows Elamipretide to specifically accumulate in the mitochondria, where it binds to cardiolipin.

This interaction stabilizes mitochondrial membranes and prevents the detachment of proteins critical for maintaining mitochondrial function [1]. Each component of Elamipretide contributes to its unique ability to protect mitochondria and improve mitochondrial function, which underpins its efficacy in treating cardiovascular and other mitochondrial-related diseases.

## 3. Mechanism of Action

### 3.1. Cardiolipin

Cardiolipin, a phospholipid found almost exclusively in the inner mitochondrial membrane, plays a crucial role in maintaining mitochondrial cristae structure and supporting the function of electron transport chain complexes [17]. During mitochondrial dysfunction, cardiolipin undergoes oxidation, leading to impaired mitochondrial bioenergetics and cell death [3]. Cardiolipin is characterized by its unique structure, which consists of a glycerol backbone with two phosphate groups and four fatty acid chains. This structure is distinct from typical phospholipids, which usually have two fatty acid chains. Cardiolipin contains four acyl chains, often consisting of two unsaturated and two saturated fatty acids. The specific fatty acids can vary, leading to different cardiolipin species [5,15].

### 3.2. Biosynthesis of Cardiolipin

Cardiolipin is primarily synthesized in the mitochondria, with the biosynthesis of the precursor phosphatidic acid occurring in the endoplasmic reticulum and outer mitochondrial membrane, followed by all subsequent steps occurring on the matrix side of the inner mitochondrial membrane [18]. Cardiolipin biosynthesis involves several key steps, including precursor formation, conversion to cardiolipin, and remodeling. Biosynthesis begins with the formation of precursor phosphatidic acid (PA), which is then converted to cytidinediphosphate-diacyl-glycerol (CDP-DAG) by the enzyme CDP-DAG synthase [19]. CDP-DAG is then converted to phosphatidylglycerol phosphate (PGP) in a rate-limiting step by the enzyme PGP synthase, followed by dephosphorylation to form phosphatidylglycerol (PG). As shown in Figure 3, the enzyme cardiolipin synthase (CLS) then catalyzes the synthesis of nascent cardiolipin from the condensation of PG and CDP-DAG [19,20].

Nascent cardiolipin is then deacylated to monolysocardiolipin (MLCL) to allow for remodeling to obtain tissue-specific acyl groups necessary for functioning [20]. The subsequent reacylation of MLCL to form mature cardiolipin is proposed to be catalyzed by enzymes such as MLCLAT1, ALCAT1, and Tafazzin [21].

### 3.3. Main Functions of Cardiolipin

Cardiolipin serves several essential functions in cellular metabolism and mitochondrial function including mitochondrial integrity and structure, energy production, apoptosis, and cell signaling. Cardiolipin is critical for maintaining the structural integrity of the inner mitochondrial membrane, influencing the formation of mitochondrial cristae, which are vital for optimal ATP production through oxidative phosphorylation [3]. Cardiolipin plays a role in mitochondrial membrane fusion and fission—processes that are essential for mitochondrial quality control and distribution within cells. Cardiolipin is also involved in the function of key enzymes of the electron transport chain (ETC), specifically by facilitating the assembly and activity of complexes I, III, and IV, thereby enhancing ATP production. Cardiolipin is involved in the intrinsic apoptotic pathway. During apoptosis, cardiolipin translocates from the inner to the outer mitochondrial membrane, where it interacts with cytochrome C, promoting its release and the subsequent activation of caspases [19,21]. Cardiolipin is implicated in several signaling pathways, influencing fundamental cellular processes including inflammation, oxidative stress response, and cell survival.

### 3.4. Clinical Relevance of Cardiolipin

Cardiolipin abnormalities have been implicated in numerous mitochondria-associated conditions including cardiovascular diseases, neurodegenerative disorders, metabolic disorders, and aging [3]. Altered cardiolipin content in heart tissue correlates with impaired cardiac function and increased susceptibility to ischemic damage [21]. Changes in cardiolipin composition can affect mitochondrial function during reperfusion, thus impacting cell survival and contributing to ischemia–reperfusion injury. Other cardiovascular diseases associated with altered cardiolipin include hypertrophic and dilated cardiomyopathies, heart failure, and Barth syndrome [17,21]. Neurodegenerative disorders such as Parkinson’s disease and Alzheimer’s disease have been correlated with altered cardiolipin metabolism. Altered cardiolipin profiles have also been observed in metabolic disorders such as diabetes and obesity, where energy metabolism and mitochondrial function may be affected. Lastly, cardiolipin levels and composition have been shown to change with age, contributing to age-related decline in mitochondrial function [3].

### 3.5. Interaction with Cardiolipin

Elamipretide localizes to the inner mitochondrial membrane and binds to cardiolipin via electrostatic interactions due to its positively charged amino acid residues. This binding stabilizes cardiolipin, preventing oxidative damage and maintaining mitochondrial membrane potential [1]. Elamipretide reduces cardiolipin peroxidation, thus preserving mitochondrial structure and function through the maintenance of cristae integrity, reduction in ROS production, and preservation of mitochondrial ATP production. If left unchecked, cardiolipin oxidation can disrupt cristae structure and promote mitochondrial dysfunction and apoptosis [5]. The stabilization of cardiolipin by Elamipretide results in improved mitochondrial function, reduced oxidative stress, and mitigated cell damage, all of which are essential factors of its therapeutic potential in diseases involving mitochondrial dysfunction.

### 3.6. Enhancement of Mitochondrial Bioenergetics

Mitochondria produce ATP through oxidative phosphorylation (OXPHOS) by the ETC. Consequently, impaired functioning of the ETC decreases ATP generation, leading to energy deficits in cells, especially in highly energetic tissues like the heart [3]. Elamipretide enhances the activity of mitochondrial respiratory complexes (I, III, and IV) by promoting their assembly and stability [5]. This enhancement facilitates more efficient electron transfer and ATP synthesis. By improving the function of the electron transport chain, Elamipretide minimizes electron leakage, a significant source of mitochondrial ROS [3]. A reduction in ROS levels prevents oxidative damage to mitochondrial components including cardiolipin [1,5]. Elamipretide plays a crucial role in restoring the mitochondrial membrane potential (ΔΨ_m_) essential for ATP synthesis. By preventing cardiolipin peroxidation and stabilizing the inner mitochondrial membrane, Elamipretide helps maintain ΔΨ_m_, critical for ATP production [5].

### 3.7. Inhibition of Mitochondrial Permeability Transition Pore (mPTP) Opening

The mitochondrial permeability transition pore (mPTP) is a channel that opens during stress (such as during ischemia–reperfusion injury), leading to mitochondrial swelling, depolarization, and the release of pro-apoptotic factors, triggering cell death [3,17]. Elamipretide reduces oxidative stress and stabilizes cardiolipin, maintaining the ΔΨ_m_, and thus inhibiting the opening of mPTP [1]. By preventing mPTP opening, Elamipretide protects against mitochondrial damage during reperfusion and reduces cell death and tissue damage in ischemic conditions [2].

### 3.8. Reduction in Mitochondrial Reactive Oxygen Species (ROS)

Excess ROS production within the mitochondria leads to oxidative damage, which is involved in the pathogenesis of cardiovascular diseases, ischemia–reperfusion injury, and heart failure [22,23,24,25,26,27,28,29,30,31,32,33,34,35,36,37]. Elamipretide reduces ROS production by improving the efficiency of the electron transport chain, thus preventing oxidative stress. High levels of ROS can damage mitochondrial DNA, proteins, and lipids, leading to impaired mitochondrial function [1]. Elamipretide improves the efficiency of the electron transport chain and reduces electron leakage, which is a major source of ROS production in mitochondria. By reducing ROS levels, Elamipretide helps protect mitochondrial lipids like cardiolipin and mitochondrial proteins from oxidative damage [3].

### 3.9. Anti-Apoptotic and Anti-Fibrotic Effects

Mitochondrial dysfunction leads to the activation of apoptotic pathways through the release of cytochrome c and the activation of caspases [2,5,38]. Over time, chronic mitochondrial damage and oxidative stress contribute to fibrosis, especially in the heart. By preserving mitochondrial function, Elamipretide prevents the release of cytochrome c and the subsequent activation of caspase-dependent apoptotic pathways [5]. This is particularly important in preventing cardiomyocyte death in heart failure and ischemic conditions. Preclinical studies have shown that Elamipretide reduces fibrosis in heart tissue by mitigating mitochondrial dysfunction and oxidative stress, both of which are key drivers of fibroblast activation and extracellular matrix deposition [17].

The primary mechanisms of action are centered around protecting mitochondrial structure and function, particularly by interacting with cardiolipin. By preventing mitochondrial dysfunction, reducing ROS production, inhibiting mPTP opening, and mitigating apoptosis and fibrosis, Elamipretide shows significant promise in the treatment of cardiovascular and other mitochondria-related diseases.

## 4. Preclinical Studies

Elamipretide has been extensively studied in preclinical animal models due to its ability to mitigate mitochondrial dysfunction and its potential therapeutic applications in cardiovascular, renal, and neurodegenerative diseases (Table 1).

### 4.1. Cardiovascular Disease Models

Mitochondrial dysfunction plays a critical role in the pathogenesis of heart failure, contributing to a discrepancy between the supply and demand of energy. Examples of such pathogenesis include post-ischemic reduction in mitochondrial ATP production or increased myocardial energetic demand from altered physiological mechanisms of the body [3]. Biopsies of human hearts have shown a significant decrease in myocardial ATP that correlates with impaired myocardial contraction and relaxation seen in heart failure. An investigation into mitochondrial abnormalities in the myocardium of dogs with chronic heart failure found the severity of mitochondrial injury to be associated with the severity of left ventricular dysfunction [39]. Cardiac mitochondrial injury in heart failure includes reduced organelle size, diminished rate of ATP synthesis, increased formation of ROS, and structural abnormalities like loss of cristae, which indicate abnormal cardiolipin [3,17]. Consequently, Elamipretide is of interest due to its ability to stabilize cardiolipin and reduce oxidative stress, thus potentially recovering cardiac mitochondrial function and improving cardiac performance.

A preclinical study using dogs with advanced heart failure found that chronic therapy with Elamipretide improved left ventricular function and prevented progressive left ventricular enlargement [17]. Acute intravenous infusion of Elamipretide was found to significantly decrease left ventricular end-systolic volume (LVESV) and significantly increase ejection fraction (EF) and stroke volume (SV), indicating beneficial effects on cardiac function. Three months of subcutaneous Elamipretide therapy was associated with significantly increased SV, EF, cardiac output (CO), and cardiac index, along with decreased left ventricular end-diastolic pressure (LVEDP), systemic vascular resistance (SVR), and end-diastolic wall stress. These improvements in left ventricular function were accompanied by normalization of heart failure biomarkers (natriuretic peptides and proinflammatory cytokines) and reversal of mitochondrial dysfunction evidenced by improved ATP synthesis and reduced ROS formation. Dogs treated with Elamipretide also displayed reduced cardiomyocyte hypertrophy and interstitial fibrosis, as well as increased capillary density, indicating management of left ventricular remodeling [17]. These results suggest that Elamipretide has the ability to stabilize mitochondria and normalize cardiolipin, thereby improving cardiac muscle performance and energy production during heart failure and preventing consequential cardiac hypertrophy and fibrosis.

Animal models of heart failure were further used to investigate the efficacy of Elamipretide on restoring skeletal muscle function [40]. Exercise intolerance is a hallmark of chronic heart failure, and it has been associated with skeletal muscle atrophy, characterized by a transition from slow-twitch type 1 (oxidative) to fast-twitch type 2 (glycolytic) muscle fibers and energetic mitochondrial abnormalities. Treatment with Elamipretide restored a near-normal type 1:type 2 muscle fiber ratio in the dogs. Elamipretide was also found to cause a dose-dependent improvement in mitochondrial function in skeletal muscle mitochondria. While the study did not investigate the effects on exercise intolerance, the results of normalized skeletal muscle morphology and improved mitochondrial function suggest potential for addressing exercise intolerance in heart failure [40].

Another study demonstrated that post-infarction chronic Elamipretide therapy improved cardiac function and prevented left ventricular remodeling in a rat myocardial infarction (MI) model. Following an MI, the heart has a decreased energy supply due to mitochondrial injury and reduced ATP synthesis. Ultimately, cardiac damage from the infarct and reduced energy supply may culminate in heart failure. Thus, improving cardiac function post-MI is a pertinent endeavor, and Elamipretide may be a useful tool to do so, as it preserved SERCA2a (depressed SERCA2a is a major factor in heart failure) and reduced cardiac fibrosis in post-MI rats [41].

### 4.2. Renal Disease Models

Elamipretide has shown promise in renal disease models, particularly in mitigating damage from ischemia–reperfusion injury and diabetic nephropathy, both of which are associated with mitochondrial dysfunction [3].

In a mouse model of ischemia–reperfusion injury, administration of Elamipretide led to a reduction in kidney injury, as demonstrated by improved mitochondrial integrity and decreased oxidative stress. The peptide accelerated the recovery of ATP levels in renal tissue, supporting its potential in acute kidney injury [3].

In murine models of diabetic nephropathy, a condition characterized by mitochondrial damage due to chronic hyperglycemia, Elamipretide protected the mitochondrial structure and restored mitochondrial function in renal cells. This helped reduce oxidative damage and fibrosis in kidney tissue, suggesting that Elamipretide might slow the progression of chronic kidney disease in diabetics [42,43].

### 4.3. Neurodegenerative Disease Models

Mitochondrial dysfunction and oxidative stress have also been implicated in neurodegenerative diseases such as Alzheimer’s disease and Parkinson’s disease. Preclinical studies have investigated Elamipretide for its neuroprotective potential.

In Alzheimer’s disease, the accumulation of amyloid-beta (Aβ) within neural mitochondria impairs the mitochondrial electron transport chain, leading to ATP reduction [44]. A study utilizing a mouse model of Alzheimer’s disease and mouse neuroblastoma (N2a) cells incubated with the amyloid-beta (Aβ) peptide found recovered mitochondrial function and increased neurite outgrowth with Elamipretide therapy, indicating protection against Aβ toxicity [45]. A separate in vitro study found that Elamipretide upregulated neural mitochondrial biogenesis against Aβ aggregation in N2a cells transfected with mutant AβPP cDNA [44]. Overall, preclinical studies show promise for Elamipretide in protecting against Aβ toxicity in Alzheimer’s disease.

Elamipretide was also evaluated in mice models of Parkinson’s disease and produced dose-dependent protection of dopaminergic neurons from oxidative damage and mitochondrial dysfunction, suggesting its potential for slowing the progression of neurodegeneration in Parkinson’s disease [46].

### 4.4. Doxorubicin-Induced Cardiotoxicity Models

Doxorubicin is a commonly used chemotherapeutic agent notorious for causing dose-dependent cardiotoxicity, causing irreversible heart damage and heart failure. The cardiotoxic effects of doxorubicin are primarily mediated through the generation of excess ROS in mitochondria, leading to oxidative damage of mitochondrial DNA, lipids (particularly cardiolipin), and proteins [3]. This results in impaired mitochondrial function, reduced ATP production, and the activation of cell death pathways (apoptosis and necrosis) in cardiomyocytes [5]. Cardiolipin-targeting mechanism makes Elamipretide particularly effective at addressing the mitochondrial damage induced by doxorubicin and several preclinical studies have demonstrated the potential of Elamipretide to mitigate doxorubicin-induced cardiotoxicity.

In a study using rat models of dilated cardiomyopathy induced by doxorubicin, adjuvant Elamipretide in addition to Sacubitril/Valsartan superiorly preserved cardiac function and LVEF, as well as reduced myocardial fibrosis. Early administration of a combination of Elamipretide and Entresto remarkably attenuated the inflammatory reaction, oxidative stress, and mitochondrial damage pathways central to the pathogenesis of Doxorubicin-induced dilated cardiomyopathy. The combination therapy also substantially reduced the levels of BNP, which is a significant biomarker for predicting pressure overload and heart failure [47]. Another study with H9c2 cells and mice models demonstrated the cardioprotective effects of Elamipretide in vitro and in vivo by reducing ROS, stabilizing the mitochondrial membrane potential, and attenuating myocardial apoptosis and fibrosis following doxorubicin treatment [48]. The ability of Elamipretide to stabilize cardiolipin, reduce ROS production, and preserve mitochondrial function has demonstrated promise in preventing the onset of cardiomyopathy and heart failure caused by doxorubicin treatment, thereby providing a pathway to reduce the cardiovascular side effects of cancer treatments.

## 5. Clinical Trials

Elamipretide has been evaluated in multiple clinical trials to determine its therapeutic potential across a range of conditions characterized by mitochondrial dysfunction. The primary endpoints of these trials generally include assessing safety, tolerability, and efficacy in improving mitochondrial function and clinical outcomes (Table 2).

### 5.1. EMBRACE-STEMI

The EMBRACE-STEMI clinical trial is a multicenter, randomized, double-blind Phase 2a study that evaluated the efficacy and safety of Elamipretide among first-time anterior ST-elevation myocardial infarction (STEMI) patients who received primary percutaneous coronary intervention (PCI) for a proximal or mid-left anterior descending (LAD) artery occlusion. The results of the trial showed that Elamipretide was not associated with a decrease in myocardial infarct size. However, the reduced incidence of heart failure within 24 h following PCI, which comprised approximately 75% of all new-onset heart failure events, was associated with Elamipretide treatment—although the incidence of heart failure after 24 h was not reduced [49].

### 5.2. PROGRESS-HF

Mitochondrial dysfunction is a critical factor in heart failure with reduced ejection fraction (HFrEF). The PROGRESS-HF trial is a randomized, double-blinded, placebo-controlled Phase 2 study that enrolled patients with stable HFrEF in 20 centers in Europe that assessed the effects of Elamipretide on left ventricular function in HFrEF patients. While improvements were observed in mitochondrial function and quality of life, the primary endpoint (a reduction in LVESV) was not achieved. Secondary analyses of EF and left ventricular end-diastolic volume (LVEDV) also showed no significant improvements [50]. A previous small, randomized, placebo-controlled trial in patients with HFrEF found that Elamipretide significantly reduced LVEDV and LVESV [56]. It is important to note that changes noted in this smaller-scale trial did not correspond with biomarker changes and had wide confidence intervals [50]. However, possibilities to explore include longer therapy duration than the four weeks tested in the PROGRESS-HF trial, since animal-model studies tested three months of Elamipretide therapy—perhaps longer therapeutic durations are required for cardiac remodeling and correction of abnormal mitochondrial dynamics in heart failure. Another theory to explore is the chance that the mitochondrial function improvement offered by Elamipretide may be insignificant compared to the effects of contemporary guideline-directed medical therapy (GDMT) or may reveal effects only during times of higher energetic demand like during exercise. A last explanation to investigate is that Elamipretide may promote left ventricular relaxation as opposed to contraction, thus accounting for the slight increase in LVEDV as assessed by cardiac MRI without a significant decrease in LVESV [50]. Overall, while primary and secondary endpoints were not achieved, trending improvements in quality of life scores with Elamipretide may be promising areas of further investigation, especially with aforementioned preclinical studies that have shown the ability of Elamipretide to improve skeletal muscle function and increase exercise tolerance in heart failure models.

### 5.3. TAZPOWER

Barth syndrome, a rare x-linked defect in the Tafazzin (TAZ) enzyme involved in cardiolipin remodeling, has been targeted by Elamipretide in the TAZPOWER trial. Barth syndrome clinically presents as cardiac left ventricular noncompaction, early-onset cardiomyopathy, intermittent neutropenia, abnormal growth, and skeletal myopathy. While the burden of morbidity and mortality is high, there are currently no disease-specific treatments. The abnormal cardiolipin content resulting from TAZ deficiency culminates in mitochondrial deficiency, and so Elamipretide was investigated as a therapy of interest. This Phase 2/3 randomized, double-blind, placebo-controlled crossover study found that daily administration of Elamipretide for a total of 48 weeks improved symptoms of Barth syndrome including significant augmentation of skeletal muscle strength and cardiac stroke volume [57]. A subsequent 168-week open-label extension study found significant improvements from baseline on 6MWT and reduced total fatigue scores associated with long-term Elamipretide therapy. Cardiac parameters, including SV, LVEDV, and LVESV, and MLCL/CL values also showed significant improvement from baseline to week 168. Overall, Elamipretide has exhibited sustained long-term tolerability and efficacy, with improved skeletal muscle and cardiac function in Barth syndrome [51]. Since Barth syndrome often appears in infants, an area requiring further investigation is the safety and efficacy of Elamipretide in younger children with Barth syndrome [57].

### 5.4. ReCLAIM

The ReCLAIM study is a randomized, double-blind, placebo-controlled, multicenter phase 2 trial evaluating the effects of Elamipretide in patients over 55 years of age with dry age-related macular degeneration (AMD). AMD is the leading cause of irreversible blindness in people aged over 50 years, primarily caused by photoreceptor dysfunction. Preclinical studies found Elamipretide prevented and corrected vision loss in diabetic mice, as well as improved survival of human retinal endothelial cells, trabecular meshwork cells, and retinal pigmented epithelial cells via reduction in oxidative stress and apoptosis. While primary endpoints of changes from baseline in low luminance best-corrected visual acuity (LL BCVA) and square root converted geographic atrophy area were not observed, Elamipretide was associated with a slowing of progressive ellipsoid zone (EZ) degradation, which is a predictor for progressive pathological changes associated with vision loss and AMD. The EZ is thought to correspond to the mitochondria-rich layer of the photoreceptors. As such, the reduction in EZ degradation may be correlated with reduced progressive photoreceptor degeneration, which is a major factor in vision loss in AMD. The results confirm the potential of Elamipretide to preserve photoreceptor function and thus slow the progression of vision loss in dry AMD [52,53,54].

### 5.5. MMPOWER-3

Primary mitochondrial myopathies (PMM) are a group of genetic disorders characterized by impaired mitochondrial oxidative phosphorylation leading to muscle weakness and fatigue. The MMPOWER-3 trial is a randomized, double-blind, placebo-controlled phase 3 study that evaluated the safety and efficacy of Elamipretide in patients with genetically confirmed PMM. While this study did not meet its primary endpoints of improvements in the six-minute walk test (6MWT), participants treated with Elamipretide did report slightly less total fatigue as assessed by the PMMSA total fatigue score. Additionally, improvement in 6MWT outcomes was observed in a subgroup of participants with nuclear DNA (nDNA) defects [55]. Recent post hoc analyses [58] revealed a beneficial effect of Elamipretide in PMM patients with replisome disorders of mitochondrial DNA (mtDNA), highlighting the importance of accounting for specific genetic subtypes in PMM clinical trials. These findings form the basis for a follow-up Phase 3 clinical trial (NuPOWER) aimed at evaluating the efficacy of Elamipretide in patients with mtDNA maintenance-related disorders.

### 5.6. Other Trials

A phase 2a trial investigated the efficacy of Elamipretide during stent revascularization in patients with atherosclerotic renal artery stenosis (ARAS). ARAS reduces renal blood flow and may induce kidney hypoxia and ischemic nephropathy. Stent revascularization often fails to recover renal function, potentially due to ischemia/reperfusion injury. In a swine model of ARAS, Elamipretide infusion during revascularization reduced oxidative stress, tubular damage, and inflammation. In this study, adjunctive Elamipretide before and during stent revascularization was associated with attenuated postprocedural hypoxia, increased renal blood flow, and improved kidney function, as estimated by cortical blood flow and glomerular filtration rate when measured 3 months later [59].

## 6. Safety Profile

Multiple clinical trials have demonstrated a favorable tolerability profile for Elamipretide. Reported side effects have been mild and transient, with the most common event being injection site reactions, including pain at the injection site, redness, and swelling. Other less common adverse events include mild to moderate headaches, dizziness, nausea, abdominal pain, and fatigue during treatment [50,51,54,55,57]. Reports of more severe side effects include rare instances of urticaria [51]. The clinical trials noted no clinically significant differences in vital signs, laboratory values, physical exam, and ECG-measured parameters, even for the 168-week open-label extension of Elamipretide for the TAZPOWER trial [51,55]. Overall, Elamipretide has been shown to be generally well-tolerated, leading to the conclusion that the benefits of Elamipretide outweigh the risks for patients with mitochondrial dysfunction-related conditions. Continued research and monitoring will provide further insights into the safety and tolerability of Elamipretide, which may facilitate its use in clinical practice for various conditions associated with mitochondrial dysfunction.

## 7. Conclusions

Elamipretide is a promising therapeutic agent targeting mitochondrial dysfunction, which underlies pathogenesis for a variety of diseases. By selectively interacting with cardiolipin, an essential component of the mitochondrial membrane, Elamipretide stabilizes mitochondrial structure and function, mitigates oxidative stress, and enhances ATP production. The unique biochemical properties of this tetrapeptide contribute to its stability, localization to the inner mitochondrial membrane, and high cellular permeability. Although preclinical and clinical studies have demonstrated the potential of Elamipretide in treating various pathologies, challenges remain in optimizing its delivery, understanding long-term effects, and expanding its applications to diverse mitochondrial-related conditions. The evidence to date supports its safety and efficacy, warranting further investigation into larger and more diverse patient populations. Future research may expand upon preclinical findings to develop clinical trials for neurodegenerative diseases, metabolic disorders, and other conditions where mitochondrial dysfunction is a core pathology. Ongoing and future clinical trials will be critical in determining the full potential of Elamipretide in clinical practice. Elamipretide represents a breakthrough in therapies targeting mitochondria, offering a novel mechanism of intervention in otherwise intractable diseases.

## Figures and Tables

**Figure 1 ijms-26-00944-f001:**
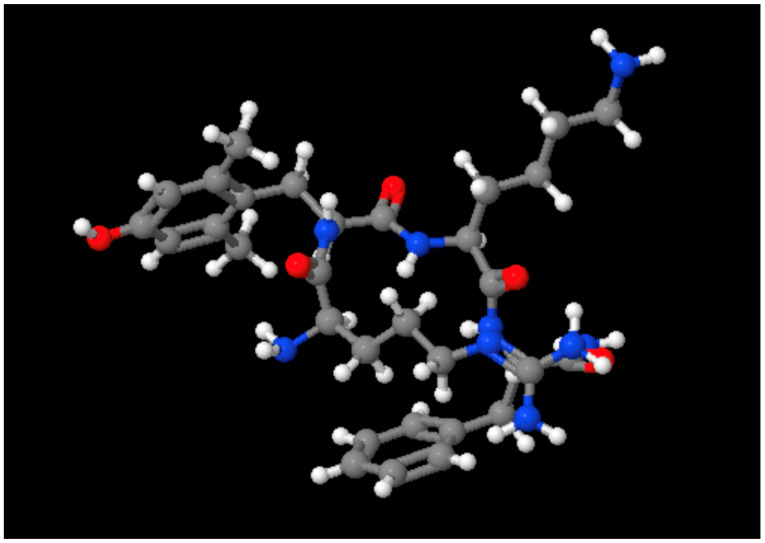
Ball and stick model of the structure of Elamipretide.

**Figure 2 ijms-26-00944-f002:**
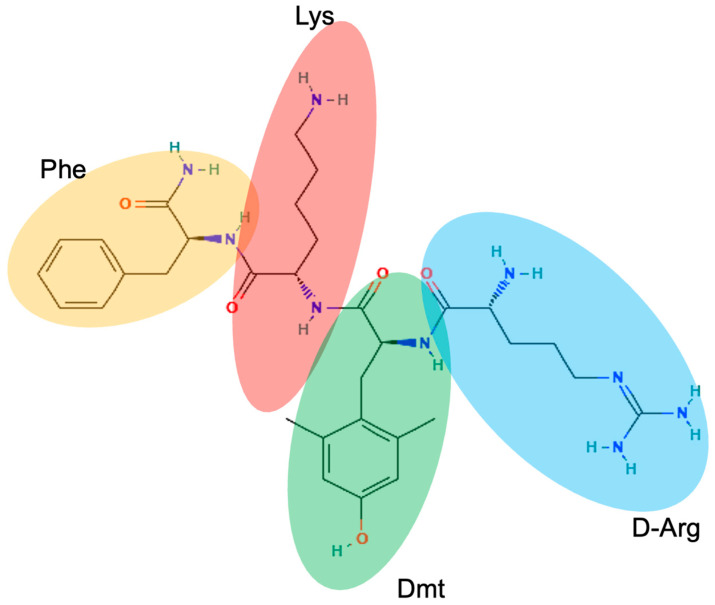
Schematic representation of the tetrapeptide with the sequence (D-Arg-Dmt-Lys-Phe-NH2) of Elamipretide.

**Figure 3 ijms-26-00944-f003:**
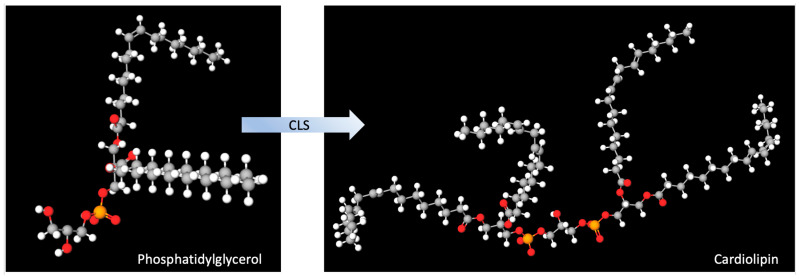
The enzyme cardiolipin synthase (CLS) catalyzes the synthesis of cardiolipin from phosphatidylglycerol.

**Table 1 ijms-26-00944-t001:** Summary of the main preclinical studies investigating the properties of Elamipretide.

Model	Design	Results	Interpretation
**Cardiovascular Disease**	Preclinical studies in animal models of heart failure.	Chronic therapy improved LV function, reduced LVESV, increased EF, stroke volume, and cardiac output, while reversing mitochondrial dysfunction, hypertrophy, and fibrosis. Skeletal muscle mitochondrial function and morphology were also restored.	The intervention stabilizes cardiolipin, improves energy production, and prevents pathological remodeling, suggesting potential for treating heart failure and post-MI cardiac dysfunction.
**Renal Disease**	Murine models of ischemia–reperfusion injury and diabetic nephropathy.	The intervention reduced oxidative stress, restored ATP levels, and protected mitochondrial structure, leading to decreased kidney injury and fibrosis.	The results highlight the ability to mitigate acute and chronic kidney injury by improving mitochondrial function, offering therapeutic potential for renal diseases.
**Neurodegenerative Disease**	Murine models of Alzheimer’s and Parkinson’s diseases.	Treatment protected against amyloid-beta toxicity in Alzheimer’s models and preserved dopaminergic neurons in Parkinson’s models, improving mitochondrial function and reducing oxidative stress.	The intervention indicates neuroprotective potential by mitigating mitochondrial damage, offering promise for slowing neurodegeneration in Alzheimer’s and Parkinson’s diseases.

**Table 2 ijms-26-00944-t002:** Main clinical trials testing Elamipretide.

Name	Design	Results	Interpretation	Ref(s).
**EMBRACE-STEMI**	Multicenter, randomized, double-blind Phase 2a study in first-time anterior STEMI patients undergoing PCI for proximal or mid-LAD artery occlusion. **Dose**: 0.05 mg/kg/h for 1 h.	No reduction in myocardial infarct size; however, a reduced incidence of chronic heart failure was observed within 24 h post-PCI.	While Elamipretide did not impact infarct size, its potential to reduce chronic heart failure warranted further investigation into its cardioprotective effects.	[49]
**PROGRESS-HF**	Randomized, double-blind, placebo-controlled Phase 2 trial assessing left ventricular function in patients with stable HFrEF. **Dose**: 4 or 40 mg.	The primary endpoint (LVESV reduction) was not met; mitochondrial function and quality of life showed improvements, though secondary cardiac parameters were unchanged.	Longer therapy duration or evaluation during exercise may better capture the benefits of Elamipretide, particularly for cardiac remodeling and mitochondrial dynamics.	[50]
**TAZPOWER**	Randomized, double-blind, placebo-controlled crossover study in Barth syndrome, followed by a 168-week open-label extension. **Dose**: 40 mg.	Significant improvements in skeletal muscle strength, cardiac stroke volume, fatigue scores, and cardiac parameters (SV, LVEDV, LVESV) were observed.	Sustained efficacy and tolerability highlight the potential of Elamipretide as a therapy for Barth syndrome, with further research needed in younger patients.	[51]
**ReCLAIM (1 and 2)**	Randomized, double-blind, placebo-controlled Phase 2 trial in patients with dry AMD. **Dose**: 40 mg.	Primary endpoints (visual acuity and geographic atrophy area) were not met, but progressive EZ degradation, a predictor of vision loss, was slowed.	Elamipretide may preserve photoreceptor function by protecting mitochondria, potentially slowing AMD progression.	[52,53,54]
**MMPOWER-3**	Randomized, double-blind, placebo-controlled Phase 3 trial in patients with genetically confirmed PMM. **Dose**: 40 mg.	Primary endpoints, including 6MWT, were not met; however, reduced fatigue was reported, particularly in participants with DNA defects.	Subgroup benefits in DNA-related PMM highlight the need for targeted trials to confirm the efficacy of Elamipretide in specific genetic variants. Further investigations are needed to explain why some endpoints were not met.	[55]

## Data Availability

Not applicable.

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
