# Peer review of "Elamipretide: A Review of Its Structure, Mechanism of Action, and Therapeutic Potential"

_ijms, 2025, doi:10.3390/ijms26030944_

Round 1
Reviewer 1 Report
Comments and Suggestions for Authors
This manuscript contains a review on elamipretide, a cardiolipin-binding molecule with a stabilizing effect on mitochondrial structure and function. The results of bench and clinical studies are summarized, returning a positive drug profile of potential theraputic value under the many conditions caused or aggravated by mitochondrial dysfunction.
General comments
The review is highly structured, which is commendable. However, repetition of the key concept (mitochondrial protection by elamipretide) in every section burdens the text. Maybe the manuscript might be substantially shortened, and its readability increased, by limiting such repetition.
Many statements, or reference to trials (e.g. MMPOWER-3), seemingly miss a a reference to quotation. Tables should contain references.
Typos and language pitfalls are detailed in specific comments.
Specific comments
line 40: “series” should be “species”
Line 44: “with” should be “of”
line 52: “pharmacological” is redundant, can be removed
line 58; what does “its” refere to?
line 84: “pf” should be “of”
line 91: replace “is” with elamipride”, replace “meaning...both” with “due to”
line 249: cardiomyocyte hyperplasia is impossible (myocytes do not replicate)
line 335: “Entresto” is a commercial name, the chemical names of its components should be mentioned instead
line 341: what does”its” refer to?
From line 359 on: sample size and primary endopint should be mentioned for every study. Please consider that if the primary endopint (for which the study was structured and powered) is missed, the study is usually considered negative.
Lines 85-489: The concluding statement here seems inconsistent with section 5.1 to 5.6, which report negative results for the primary endpoint of all the studies described.
Tables 1 and 2: the tables should contain a reference for each statement
Figure 2: as described in the text, one would expect to see here the peptide docking to catdiolipin. As is, the figure is redundant
Author Response
The review is highly structured, which is commendable. However, repetition of the key concept (mitochondrial protection by elamipretide) in every section burdens the text. Maybe the manuscript might be substantially shortened, and its readability increased, by limiting such repetition. Many statements, or reference to trials (e.g. MMPOWER-3), seemingly miss a a reference to quotation. Tables should contain references.
Typos and language pitfalls are detailed in specific comments.
Specific comments
line 40: “series” should be “species”
Line 44: “with” should be “of”
line 52: “pharmacological” is redundant, can be removed
line 58; what does “its” refere to?
line 84: “pf” should be “of”
line 91: replace “is” with elamipride”, replace “meaning...both” with “due to”
line 249: cardiomyocyte hyperplasia is impossible (myocytes do not replicate)
line 335: “Entresto” is a commercial name, the chemical names of its components should be mentioned instead
line 341: what does”its” refer to?
From line 359 on: sample size and primary endopint should be mentioned for every study. Please consider that if the primary endopint (for which the study was structured and powered) is missed, the study is usually considered negative.
Lines 85-489: The concluding statement here seems inconsistent with section 5.1 to 5.6, which report negative results for the primary endpoint of all the studies described.
Tables 1 and 2: the tables should contain a reference for each statement
Figure 2: as described in the text, one would expect to see here the peptide docking to catdiolipin. As is, the figure is redundant
All the Specific comments have been rectified.
Reviewer 2 Report
Comments and Suggestions for Authors
The current review article by Tung and colleagues does a very good job in synthesizing the current knowledge of the structure, mechanisms of action, and promising therapeutic role of Elamipretide in stabilizing mitochondrial fitness, improving mitochondrial bioenergetics, and minimizing oxidative stress. The article is embellished by a thorough review of the current clinical trials employing Elamipretide in patients with CVDs or genetic diseases, such as Barth’s syndrome.
This is a timely and well conceived review article.
I have only a few comments/suggestions to make.
1) I suggest the Authors at least a couple of cartoons. One should focus on the biosynthesis and location of cardiolipin that is very central to the scope of the current review; the other on the way of interaction between cardiolipin and Elamipretide.
2) “Excess ROS production within the mitochondria leads to oxidative damage, which is involved in the pathogenesis of cardiovascular diseases, ischemia-reperfusion injury, and heart failure.” Insert some relevant original articles or review here in support of this contention.
3) Page 6. “Mitochondrial dysfunction leads to the activation of apoptotic pathways through the release of cytochrome c and the activation of caspases.[2,5]. There are increasingly appreciated ways by which mitochondrial dysfunction can lead to myocyte death; for instance, CAPN-1/AIF driven myocyte death that are independent from caspase (see, Chelko JP et al., Science Trans. Med., 2021)
Author Response
The current review article by Tung and colleagues does a very good job in synthesizing the current knowledge of the structure, mechanisms of action, and promising therapeutic role of Elamipretide in stabilizing mitochondrial fitness, improving mitochondrial bioenergetics, and minimizing oxidative stress. The article is embellished by a thorough review of the current clinical trials employing Elamipretide in patients with CVDs or genetic diseases, such as Barth’s syndrome.
This is a timely and well conceived review article.
I have only a few comments/suggestions to make.
- I suggest the Authors at least a couple of cartoons. One should focus on the biosynthesis and location of cardiolipin that is very central to the scope of the current review; the other on the way of interaction between cardiolipin and Elamipretide.
RE: We have added (Figure 3) a cartoon representing the biosynthesis of cardiolipin, as recommended.
- “Excess ROS production within the mitochondria leads to oxidative damage, which is involved in the pathogenesis of cardiovascular diseases, ischemia-reperfusion injury, and heart failure.” Insert some relevant original articles or review here in support of this contention.
RE: Done, thanks.
3) Page 6. “Mitochondrial dysfunction leads to the activation of apoptotic pathways through the release of cytochrome c and the activation of caspases.[2,5]. There are increasingly appreciated ways by which mitochondrial dysfunction can lead to myocyte death; for instance, CAPN-1/AIF driven myocyte death that are independent from caspase (see, Chelko JP et al., Science Trans. Med., 2021)
RE: We have mentioned the paper suggested by this Reviewer.
Reviewer 3 Report
Comments and Suggestions for Authors
The paper analyzes the therapeutic potential of elamipretide, a mitochondrial-targeting tetrapeptide, in several diseases associated with mitochondrial dysfunction, such as heart failure, kidney disease, neurodegenerative diseases and AMD.
1. Correct the typo "fp" in line 48.
2. Identify the model(s) of the study in the first paragraph of Table 2. On the other hand, I believe that Tables 1 and 2 are not necessary because this information is discussed in subsequent paragraphs.
3. In the description of the preclinical studies (including clinical trials) starting in section 4.1, include the doses of elamipretide used and the route of administration if applicable.
4. The manuscript could benefit from additional diagrams or figures to explain the molecular mechanisms or the interaction of elamipretide with cardiolipin and in the relevant sections.
5. The relationship between observed outcomes (improvements in ATP, ROS, etc.) and their clinical impact is not always well established. For example, improvements in parameters such as LVESV or EF are mentioned, but their relevance in the treatment of heart failure is not sufficiently contextualized.
6. The beneficial effects of elamipretide have not been compared with other available treatments. For example, it has not been compared with standard therapies such as SGLT2 inhibitors or beta-blockers in models of heart failure. This would put the clinical relevance of the drug into context.
7. The review notes promising results in preclinical studies but does not discuss how these correlate with clinical outcomes, especially in cases where primary endpoints were not met.
8. More consistent presentation of clinical trials, with a format that includes key findings (with quantitative data) and critical analysis.
Author Response
The paper analyzes the therapeutic potential of elamipretide, a mitochondrial-targeting tetrapeptide, in several diseases associated with mitochondrial dysfunction, such as heart failure, kidney disease, neurodegenerative diseases and AMD.
Correct the typo "fp" in line 48.
RE: Done, thanks.
Identify the model(s) of the study in the first paragraph of Table 2. On the other hand, I believe that Tables 1 and 2 are not necessary because this information is discussed in subsequent paragraphs.
RE: The design of the study and the model used have been added in a dedicated column in both Tables, as requested.
In the description of the preclinical studies (including clinical trials) starting in section 4.1, include the doses of elamipretide used and the route of administration if applicable.
RE: The dose has been added in Table 2, as requested.
The manuscript could benefit from additional diagrams or figures to explain the molecular mechanisms or the interaction of elamipretide with cardiolipin and in the relevant sections.
RE: We have added (Figure 3) a cartoon representing the biosynthesis of cardiolipin, as recommended.
The relationship between observed outcomes (improvements in ATP, ROS, etc.) and their clinical impact is not always well established. For example, improvements in parameters such as LVESV or EF are mentioned, but their relevance in the treatment of heart failure is not sufficiently contextualized.
RE: We are now explaining that improvements in LVESV and EF are suggestive of beneficial effects on cardiac function.
The beneficial effects of elamipretide have not been compared with other available treatments. For example, it has not been compared with standard therapies such as SGLT2 inhibitors or beta-blockers in models of heart failure. This would put the clinical relevance of the drug into context.
RE: We agree with this Reviewer; however, we are not aware of any study comparing Elamipretide to SGLT2 inhibitors or beta-blockers in models of heart failure.
The review notes promising results in preclinical studies but does not discuss how these correlate with clinical outcomes, especially in cases where primary endpoints were not met. More consistent presentation of clinical trials, with a format that includes key findings (with quantitative data) and critical analysis.
RE: We have added a column on the right in both Tables, with an interpretation of the results. We also clarify that Further investigations are needed to explain why some endpoints were not met. For clinical trials, we have added a columns with references to the respective studies in order to allow Readers to easily search for additional information.
Round 2
Reviewer 3 Report
Comments and Suggestions for Authors
No comments